# Biological Activity of Oleanolic Acid Derivatives HIMOXOL and Br-HIMOLID in Breast Cancer Cells Is Mediated by ER and EGFR

**DOI:** 10.3390/ijms24065099

**Published:** 2023-03-07

**Authors:** Natalia Lisiak, Patrycja Dzikowska, Urszula Wisniewska, Mariusz Kaczmarek, Barbara Bednarczyk-Cwynar, Lucjusz Zaprutko, Blazej Rubis

**Affiliations:** 1Department of Clinical Chemistry and Molecular Diagnostics, Poznan University of Medical Sciences, Rokietnicka 3 St., 60-806 Poznan, Poland; 2Department of Cancer Immunology, Chair of Medical Biotechnology, Poznan University of Medical Sciences, Garbary 15 St., 61-866 Poznan, Poland; 3Department of Organic Chemistry, Poznan University of Medical Sciences, Grunwaldzka 6 St., 60-780 Poznan, Poland

**Keywords:** breast cancer, oleanolic acid, HIMOXOL, Br-HIMOLID, ER, EGFR

## Abstract

Breast cancer is one of the most frequently observed malignancies worldwide and represents a heterogeneous group of cancers. For this reason, it is crucial to properly diagnose every single case so a specific and efficient therapy can be adjusted. One of the most critical diagnostic parameters evaluated in cancer tissue is the status of the estrogen receptor (ER) and epidermal growth factor receptor (EGFR). Interestingly, the expression of the indicated receptors may be used in a personalized therapy approach. Importantly, the promising role of phytochemicals in the modulation of pathways controlled by ER and EGFR was also demonstrated in several types of cancer. One such biologically active compound is oleanolic acid, but due to poor water solubility and cell membrane permeability that limits its use, alternative derivative compounds were developed. These are HIMOXOL and Br-HIMOLID, which were demonstrated to be capable of inducing apoptosis and autophagy or diminishing the migratory and invasive potential of breast cancer cells in vitro. In our study, we revealed that proliferation, cell cycle, apoptosis, autophagy, and also the migratory potential of HIMOXOL and Br-HIMOLID in breast cancer cells are mediated by ER (MCF7) and EGFR (MDA-MB-231) receptors. These observations make the studied compounds interesting in the context of anticancer strategies.

## 1. Introduction

In 2020, there were 2.3 million women diagnosed with breast cancer, which caused 685,000 deaths globally. Referring to the 5 years from 2016–2020, there were 7.8 million breast cancer patients, making it the world’s most prevalent cancer type [1]. The cause of most breast cancer cases is associated with older age, which is believed to be the most important risk factor. Other crucial factors are also recognized: certain gene mutations (mainly *BRCA1*, *BRCA2,* or *TP53*), family history of breast cancer (especially in young patients), early menarche, late menopause age, late age of the first childbirth, long-term hormonal contraception, and hormone replacement therapy [2].

Breast cancer refers to a heterogeneous group of cancers with different characteristics including histology parameters, the pattern of gene expression, metastatic potential, and prognosis [3]. Five molecular subtypes of breast cancer are defined: (i) luminal A, (ii) luminal B, (iii) epidermal growth factor receptor-positive (HER2+), (iv) triple-negative breast cancer (TNBC), and (v) seminormal [4]. Most of the diagnosed breast cancer cases represent the estrogen-receptor-positive (ER+), luminal A subtype (70%) [5]. However, the most metastatic and associated with the worst prognosis cases are represented by the TNBC subtype (20%) [6,7]. The diversity of biological and genetic features within breast cancer is a major diagnostic and therapeutic challenge. One of the most critical classification parameters that provide a properly adjusted therapeutic approach are estrogen receptors which can play the role of transcription factors [8]. Estrogen receptors are members of the superfamily of nuclear receptors that mediate the pathways associated with steroid hormones, thyroid hormones, retinoids, and vitamin D, as well as numerous orphan receptors [9]. Given the wide spectrum of functions of ERs, the dysregulation of their pathways contributes to the development of several diseases, including cardiovascular diseases, osteoporosis, endometrial and ovarian cancers, and hormone-dependent breast cancer [10]. Selected receptors, modulated by respective hormones, control the transcription of genes that mainly code for proteins responsible for promoting the survival, proliferation, and growth of cancer cells, as well as angiogenesis [9,11].

In general, two main types of estrogen receptors can be distinguished: ERα and ERβ (isoforms 1, 2, 4, and 5) [12]. In breast cancer, the presence of the ERα receptors is linked with tumor promotion, while the role of ERβ in tumor progression is not yet well understood [13]. Specifically, estradiol shows higher affinity to the ERα receptor than to ERβ, which can mediate the effect on cellular functions [14]. It is known, however, that ERβ, unlike ERα, acts as an oncosuppressor, by antagonizing hormone-induced carcinogenesis and inhibiting growth and oncogenic functions of cancer cells in luminal-like breast cancers [13]. Thus, estrogen receptor profiling is one of the main parameters that provide efficient therapy adjustment. Consequently, endocrine therapy aims to shut off estrogen signaling in ERα-positive breast cancer cells to decrease cancer cell proliferation and/or to induce cancer cell death [15].

Another critical element in therapy planning is the epidermal growth factor receptor, EGFR. The EGFR, also named ErbB-1 and Her1, is a member of the ErbB family which includes EGFR (ErbB-1) and three other members, i.e., HER2 (ErbB-2), HER3 (ErbB-3), and HER4 (ErbB-4) [16]. Approximately half of the cases of triple-negative breast cancer (TNBC) and inflammatory breast cancer (IBC) overexpress EGFR [17]. It is usually associated with more aggressive cases [18], large tumor size, poor differentiation, and poor clinical outcomes [19]. EGFR is a transmembrane receptor that mediates the tyrosine kinase signaling pathway to carry the extracellular signals inside the cell and alter numerous target genes [20]. This receptor is one of the first identified important targets of the antitumor agents including gefitinib, cetuximab, and lapatinib [21,22]. However, the efficacy of this strategy in breast cancer has been disappointing due to the occurrence of a drug resistance phenomenon [23,24]. Thus, novel strategies are being constantly developed, and it seems that some naturally derived compounds that mediate EGF-receptor-associated pathways may constitute an interesting alternative (or supplementation) to current strategies [25]. Many preclinical and epidemiological reports showed the promising role of phytochemicals against several types of cancer. One of them is oleanolic acid, a common pentacyclic triterpenoid, mainly found in olive oil, as well as several plant species. It is a potent inhibitor of cellular inflammation and a well-known inducer of phase 2 xenobiotic biotransformation enzymes [26,27]. It also reveals, among other things, anticancer properties, possibly via PI3K/Akt/mTOR, NFκB/p53, MMPs, and EGFR signaling pathways [28]. However, its poor low-water solubility and poor permeability limit its use [29]. Thus, novel derivatives, modified with novel moieties that provide higher biological potential, are being developed. The compounds with such features are HIMOXOL and Br-HIMOLID, oleanolic acid semisynthetic derivatives with proven proapoptotic, proautophagic, anti-migratory, and anti-invasive properties in breast cancer cells in vitro [30,31,32,33]. Further evaluation of the biological potential of oleanolic acid derivatives may reveal the mechanism of their action within individual cancer types. Consequently, it may contribute to the development of more efficient anticancer strategies and potential association with pathways mediated by estrogen or epidermal growth factor receptors.

## 2. Results

### 2.1. Optimization of Receptor Blocker Concentration

First, the concentration optimization of specific blockers of estrogen (in MCF7) and epidermal growth factor (in MDA-MB-231) receptors in cancer cells was performed. Estrogen receptors targeted in MCF7 cells with fulvestrant (0.5 µM, 1 µM, or 2 µM for 24 h) led to a significant reduction in the ER levels in all applied concentrations, relative to untreated control cells (more than 80% reduction after 24 h; Figure 1). Similarly, treatment of MDA-MB-231 cells with the EGFR inhibitor gefitinib (10 µM, 20 µM, or 30 µM for 72 h) led to a significant reduction in the EGF level at the concentrations of 20 and 30 µM (60 and more than 80% reduction relative to control, respectively; Figure 1).

### 2.2. Contribution of ER and EGFR to the Response of MCF7 and MDA-MB-231 Cells to OA or Its Derivatives

To verify if blocking the ER or EGFR in breast cancer cells (MCF7 and MDA-MB-231, respectively) modulates their response to OA or its derivatives, an MTT assay was performed (24 h). Comparison of the effect of oleanolic acid in MCF7s that were ER-positive (MCF7/ER+) or ER-negative (MCF7/ER−) revealed a concentration-dependent effect and reduced viability of cells in all concentration ranges. However, most importantly, the survival of MCF7 cells was significantly higher after blocking ER (MCF7/ER−, the concentration of fulvestrant 0.5 µM; Figure 2) than in MCF7/ER+ cells. For the concentrations applied, 2.5, 5, 10, 15, 25, or 50 µM, it was 100 vs. 95%, 100 vs. 90%, 100 vs. 80%, 60 vs. 50%, 50 vs. 40%, and 30 vs. 5%, respectively. Interestingly, the MTT test showed higher cytotoxicity of HIMOXOL than OA in MCF7 cells in both cell types (MCF7/ER− and MCF7/ER+ cells). Similarly, in both cell types (ER+ and ER−) a significant survival decrease was observed when concentrations 2.5 µM and higher were applied. Moreover, MCF7/ER− cells (fulvestrant 0.5 µM) appeared more sensitive to the compound at the concentrations of 2.5 and 5 µM than MCF7/ER+ cells (survival rate 50 vs. 40% and 40 vs. 30%, respectively). Again, MCF7/ER− cells treated with Br-HIMOLID were shown to be more sensitive to the compound at the concentration range of 15–50 µM than MCF7/ER+ cells (survival rates 40 vs. 30%, 30 vs. 20%, and 20 vs. 10%, respectively) (Figure. 2).

Treatment of MDA-MB-231 cells with OA showed no significant effect in EGFR+ cells, while in MDA-MB-231/EGFR− cells a significant decrease in survival was observed at the concentration range 15–50 µM (decrease in cell viability from 40 up to 80%, relative to control cells) (Figure 2). Treatment of MDA-MB-231 cells with HIMOXOL showed higher cytotoxic properties of the compound than OA and revealed higher viability inhibition in MDA-MB-231/EGFR− than in MDA-MB-231/EGFR+ cells in the range of concentrations: 2.5, 5, 10, 15, 25, and 50 µM (20 vs. 0%; 60 vs. 5%; 95 vs. 10%; 100 vs. 10%; 100 vs. 40%; 100 vs. 70%; and 100 vs. 95%, respectively). When MDA-MB-231 cells were treated with Br-HIMOLID, a higher cytotoxic effect was observed in MDA-MB-231/EGFR− than in MDA-MB-231/EGFR+ cells. Specifically, the survival of EGFR+ cells was not significantly altered, while survival of EGFR− cells was reduced by 40, 60, 80, 90, or 100% (concentrations of 2.5, 5, 10, 15, 25, and 50 µM, respectively; Figure 2).

The IC_50_ values (Table 1) revealed that (i) modification of OA with hydroxyimino and lactone moieties (HIMOXOL and Br-HIMOLID, respectively) provoked a higher cytotoxicity effect in studied breast cancer cells, and (ii) blocking ER or EGFR receptors in breast cancer cells (MCF7/ER− and MDA-MB-231/EGFR− cells, respectively) decreased the sensitivity of both studied derivatives relative to the maternal compound, OA, while it provoked increased cytotoxicity of the two derivatives.

Specifically, in MCF7/ER− cells, the IC_50_ for oleanolic acid was almost 3-fold higher than in MCF7/ER+ cells (31.7 µM vs. 12.7 µM). In turn, cell treatment with HIMOXOL or Br-HIMOLID revealed an opposite effect, and 2-fold lower IC_50_ values in ER− cells in comparison to ER+ cells were observed (1.63 µM vs. 3.2 µM for HIMOXOL, and 3.53 µM vs. 6.58 µM for Br-HIMOLID). In MDA-MB-231/EGFR− cells the IC_50_ values for all of the compounds were significantly lower than in MDA-MB-231/EGFR+ cells (Table 1) (22.47 µM vs. >50 µM, 1.72 µM vs. 21 µM, 3.4 µM vs. >50 µM, respectively).

### 2.3. Contribution of ER and EGFR to the Genotoxic Effect of OA or Its Derivatives in MCF7 and MDA-MB-231 Cells

The clonogenic assay allows for the observation of cells’ potential to survive for a longer period after treatment of cells with studied compounds [34]. In this study, MCF7/ER− cells treated with OA, HIMOXOL, or Br-HIMOLID were more resistant to all of the compounds than ER+ cells (Figure 3A). In turn, MDA-MB-231/EGFR− cells treated with OA showed significantly decreased colony-forming ability in comparison to EGFR+ cells. However, HIMOXOL or Br-HIMOLID treatment revealed the opposite effect (Figure 3B). As demonstrated, the colony-forming ability was significantly diminished at the concentrations corresponding to 0.5 × IC_50_, 1 × IC_50,_ and 1.5 × IC_50_, i.e., 90 vs. 50%, 60 vs. 5%, and 60 vs. 5% (MDA-MB-231/EGFR− vs. MDA-MB-231/EGFR+, respectively) for HIMOXOL, and 75 vs. 85% (with no significance), 95 vs. 60%, and 95 vs. 55% for Br-HIMOLID (MDA-MB-231/EGFR− vs. MDA-MB-231/EGFR+, respectively).

### 2.4. Contribution of ER and EGFR to the Cell Cycle Distribution in MCF7/ER+/− and MDA-MB-231/EGFR+/− Breast Cancer Cells Exposed to OA or Its Derivatives

Treatment of MCF7/ER− with OA, HIMOXOL, or Br-HIMOLID led to an increased number of cells in G0/G1 phase cells in the whole range of concentration (i.e., 0.5 × IC_50_, 1 × IC_50_, and 1.5 × IC_50_) when compared to MCF7/ER+ cells. This effect was accompanied by a significant decrease in cell population in S and G2/M phases and this effect was higher in comparison to MCF7/ER+ for all of the studied compounds. None of the tested compounds revealed proapoptotic properties in the studied cells.

When MDA-MB-231/EGFR− cells were treated with OA or its two derivatives, a decreased number of cells in the G2/M phase, in comparison to MDA-MB-231/EGFR+ cells, was observed in the whole concentration range. To emphasize, the proapoptotic effect of HIMOXOL was observed only in MDA-MB-231/EGFR+ cells in a concentration corresponding to 1 × IC_50_ (*p* < 0.05, relative to MDA-MB-231/EGFR− cells) (Figure 4).

### 2.5. Evaluation and Verification of Autophagy Pathway

Since the lowered viability of studied cells after treatment with OA or its derivatives was not associated with apoptosis in cell cycle analysis, we decided to evaluate the potential effect of studied compounds on autophagy. Consequently, the levels of autophagy-associated proteins, i.e., mTOR, BECN1, LC3-II, and p62, were evaluated after treatment with studied compounds in MCF7/ER- and MDA-MB-231/EGFR− cells.

In MCF7/ER− cells we observed a significant decrease in the level of mTOR when cells were treated with the highest concentration (1.5 × IC_50_) of OA, while the two other concentrations (0.5 and 1 × IC_50_) did not show any significant alteration, relative to control cells. When cells were treated with HIMOXOL or Br-HIMOLID, a significant decrease in mTOR was observed in all concentration ranges (i.e., 0.5, 1, or 1.5 × IC_50_). Evaluation of the BECN1 protein in MCF7/ER− cells revealed no alteration of this protein in cells treated with OA (0.5 or 1 × IC_50_), while administration of the highest concentration (1.5 × IC_50_) provoked a significant decrease in BECN1. Assessment of the p62 protein revealed no significant effect in MCF7/ER− treated with OA or HIMOXOL, while incubation with Br-HIMOLID provoked a concentration-dependent increase in this protein up to 145% (1.5 × IC_50_), relative to control cells. A very similar effect was observed when the LC3-II protein was evaluated (55% upregulation at 1.5 × IC_50_ Br-HIMOLID only) (Figure 5A).

Evaluation of mTOR in MDA-MB-231/EGFR− cells revealed no alterations when OA was applied. In cells treated with HIMOXOL, a significant reduction in mTOR was observed only at the concentration 1.5 × IC_50_, while Br-HIMOLID treatment led to a significant decrease in target protein in all concentrations up to 30%, relative to control cells. Interestingly, assessment of the BECN1 protein showed no alterations in any sample, except for positive control, i.e., treated with rapamycin (conc. 100 nM). In turn, evaluation of LC3-II protein showed a decrease in the protein when OA was applied with a significant effect at 1.5 × IC_50_ (reduction by ca. 30%), and also in the lowest applied concentration of HIMOXOL (0.5 × IC_50_). However, Br-HIMOLID in the highest concentration revealed a significant increase in the LC3-II protein level (up to 140%). The level of p62 in MDA-MB-231/EGFR− was not altered by any applied compound (Figure 5B).

### 2.6. Contribution of ER and EGFR to Migration Potential of OA and Its Derivatives in Breast Cancer Cells

To investigate the contribution of ER and EGFR receptors to alterations in the migration of MCF7/ER− and MDA-MB-231/EGFR− breast cancer cells, a wound healing assay was performed. In this experiment, cells were treated with a concentration corresponding to 0.5 × IC_50_ of OA, HIMOXOL, or Br-HIMOLID. In MCF7/ER− cells treated with OA, only a slight decrease in the number of cells migrated to the gap was reported (ca. 10%), relative to control, untreated cells. After treatment of cells with HIMOXOL or Br-HIMOLID, a 20% reduction in the number of migrated cells was observed.

In MDA-MB-231/EGFR− cells treated with OA, a 20% decrease in the number of cells was observed, relative to control, untreated cells. However, a weaker effect was observed after treatment either with HIMOXOL or Br-HIMOLID (a 10% decrease in the number of cells migrated to the gap) (Figure 6). Importantly, we also observed a significant difference in the basal migratory potential between the two studied cell lines. It reflects literature data indicating MDA-MB-231 cells as more invasive, which refers to their migration potential as well [35].

### 2.7. Contribution of OA and Its Derivatives to the Adhesion and Migration of ER− and EGFR− Deprived Breast Cancer Cells

To verify the mechanism of the anti-migratory potential of OA and its derivatives in MCF7/ER− and MDA-MB-231/EGFR− breast cancer cells, the effect of the compounds on the protein level (WB) of integrin β1, Tyr397 FAK, total FAK, and paxillin were analyzed.

Treatment of MCF7/ER− cells with OA did not alter integrin β1 and total FAK protein levels, however, it reduced Tyr397 FAK and significantly increased the paxillin protein level by 40%. Similarly, HIMOXOL did not reveal any significant effect on integrin β1, however, it reduced Tyr397 FAK and total FAK (reduction by 50%), as well as paxillin level (reduction by 20%). In the case of Br-HIMOLID, decreased integrin β1 (20%), Tyr397FAK (total reduction), FAK (50%), and no alteration in the paxillin level were observed (Figure 7).

Treatment of MDA-MB-231/EGFR− cells with OA did not alter integrin β1 and total FAK, however, it reduced Tyr397 FAK (total reduction) and paxillin levels (by 20%). In cells treated with HIMOXOL, reductions in Tyr397 FAK, total FAK (by 20%), and paxillin (by 40%) were observed. In turn, Br-HIMOLID decreased Tyr397 FAK (total reduction) and paxillin (by 90%) but did not significantly alter integrin β1 and FAK proteins, in comparison to control, untreated cells (Figure 7).

## 3. Discussion

The main challenge in current oncology is the ever-increasing number of cases of malignant neoplasms as well as the limited specificity of therapeutic strategies that cause serious side effects. Therefore, it is important to identify novel, natural, or chemically modified compounds that would show improved efficacy accompanied by attenuated negative adverse effects. Such compounds should show high cytotoxicity to neoplastic cells and provide better patient outcomes. It seems that good candidate compounds that meet those expectations can be derived from pentacyclic triterpenes [36]. Their pharmacological effects have been studied and documented over a wide spectrum. Oleanolic acid and its glycosides show a broad and multidirectional action, but not in all cases is this action is effective enough to be used in clinical practice [37]. Therefore, the parent compound (oleanolic acid) was chemically modified, and its synthetic derivatives were investigated to verify their biological potential in many aspects [38,39]. Thus, novel derivatives that show higher anti-inflammatory and anticancer efficacy were developed: (i) imidazole of 2-cyano-3,12-dioxoolean-1,9 (11) -dien-28-oic acid (CDDO-Im), (ii) a methyl derivative of 2-cyano-3.12 -dioxoolean-1,9 (11) -dien-28-oic (CDDO-Me), and (iii) 2-cyano-3,12-dioxooleic-1.9 (11) -dien-28-oic acid (CDDO) [40]. Particularly, they were shown to inhibit NFκB and mTOR, thus leading to apoptotic pathway induction [40]. Specifically, CDDO has indicated the strongest anti-inflammatory OA derivative that acted via inhibition of iNOS and COX2 [41]. In turn, the activity of CDDO-Me is based on the induction of autophagy in hematological neoplasms, while minimizing the incidence of side effects with the accompanying cardioprotective effect [40]. CDDOs also show potent antitumor activity, however, they lack selectivity for tumor cells, which causes serious side effects [40]. Altogether, selected OA derivatives significantly diminish the viability of cancer cells, but improved efficacy and specificity are required [42].

### 3.1. OA Derivatives in Cancer Targeting

The activity of oleanolic acid derivatives has already been demonstrated in numerous cancers, including lung cancer, glioma, multiple myeloma, bone sarcoma, prostate cancer, and breast cancer [43]. It was demonstrated that these compounds could act on various levels by inhibiting the growth of neoplasms. Namely, they prevent metastasis of malignant neoplasms by inhibiting angiogenesis, reducing cell proliferation, inducing tumor cell apoptosis, and/or inducing autophagy [29]. Promising derivatives of oleanolic acid with already proven anticancer potential are HIMOXOL (3-hydroxoimino-11-oxo-12-en-28-oic acid methyl ester) and DIOXOL (3,11-dioxo-12-en-28-oic acid methyl ester). These synthetic compounds were capable of modulating multi-drug resistance in CCRF-ADR5000 and CCRF-VCR1000 leukemia cells [44].

### 3.2. The Mechanistic Aspect of OA and Its Derivatives

In the studies carried out by Lisiak et al., it was shown that in MCF7 (ER+/PR+/EGFR−) and MDA-MB-231 (ER−/PR−/EGFR+) cells, two modifications of the OA structure, HIMOXOL and Br-HIMOLID, provoked higher toxicity than the parent compound itself, oleanolic acid [30,31,32]. In addition, it was demonstrated that in both cell lines, both derivatives showed the ability to induce autophagy. Additionally, HIMOXOL induced apoptosis in MDA-MB-231 cells. The mechanism of this activity involved p38 and JNK MAP kinases and the NFκB/p53 signaling pathway. However, the participation of any receptors (ER or EGFR) in this activity was not reported. Available data reveal that triterpenoids present an antagonistic activity against ERα, e.g., a compound isolated from *Schisandra glaucescens* Diels (cycloartane triterpenoids) [45]. Moreover, CDDO-Im was shown to effectively block the EGFR/signal transducer and activator of transcription 3 (STAT3)/Sox-2 signaling pathway and consequently metastasis of breast cancer [39].

Of note is that all studies performed so far were carried out using a medium with phenol red (which shows estrogenic activity [46]) supplemented with fetal bovine serum (which contains, among other things, hormones that modulate cell proliferation) [47]. Thus, all the studies were performed in conditions providing some modulatory factors affecting observed results. Therefore, to verify the contribution of estrogens and the ER pathway to the response mechanism of breast cancer cells to OA and its derivatives, we performed experiments with MCF7/ER− cells (application of ER inhibitor fulvestrant) and used a cell culture medium devoid of phenol red and charcoal-stripped FBS serum (devoid of growth hormones and cytokines). As demonstrated, the elimination of the estrogen-receptor-associated pathway (MCF7/ER− cells) led to reduced cytotoxicity of oleanolic acid relative to the effect observed in control MCF7/ER+ cells. Interestingly, the application of OA semisynthetic derivatives in MCF7/ER− cells provoked the opposite effect and led to increased sensitivity of these cells to both compounds, i.e., HIMOXOL and Br-HIMOLID. However, a clonogenic assay performed in MCF7/ER− cells subjected to all the compounds, OA, and its derivatives showed a higher survival potential of studied cells. The observed higher survival potential in ER cells may be associated with fulvestrant activity, which is known for its antiestrogenic properties. Available data showed that antiestrogens at low concentrations stimulate proliferation weakly, but they show no stimulation at high concentrations where they fully inhibit estrogen-stimulated proliferation [48].

When OA, HIMOXOL, or Br-HIMOLID were applied in MDA-MB-231 cells (MTT assay) that were treated with the EGFR inhibitor, a lower survival rate was observed than in MDA-MB-231/EGFR+ cells. In turn, in the clonogenic assay, we observed an opposite effect of HIMOXOL and Br-HIMOLID and a higher proliferation rate of MDA-MB-231/EGFR− cells than in MDA-MB-231/EGFR+ cells. It was accompanied by no alterations in the cell cycle analysis of EGFR− cells and diminished apoptosis induced by HIMOXOL in the concentration of 1 × IC50, as observed earlier in EGFR+ cells [30].

Even if the results observed in the MTT and clonogenic assays do not correspond in the whole concentration range, it must be noted that MTT evaluates a short-term metabolic effect, while clonogenic assay can assess the genotoxic effect in a longer time perspective. In both MCF7/ER− and MDA-MB-231/EGFR− cells, the colony formation potential after treatment with HIMOXOL and Br-HIMOLID was higher than in MCF7/ER+ and MDA-MB-231/EGFR+ cells, respectively. However, this potential was also higher for MCF7/ER− cells treated with OA, but lower in MDA-MB-231/EGFR− in comparison to cells overexpressing EGFR (MDA-MB-231/EGFR+). This may suggest that the biological activity of both compounds is associated with pathways mediated by both receptors (ER and EGFR, respectively). Similar suggestions were raised by Xie et al., who showed that ERα was the target for OA, and OA upregulated miR-503 expression through ERα in RAW264.7 cells (macrophage-like cells) [49]. Moreover, the OA analog, K73-03, was used as an effective anticancer agent that acted via targeting EGFR in pancreatic ASPC-1 cancer cells [50].

Cell cycle analysis revealed no significant alterations in cell distribution after exposition to any of the studied compounds in cells with either blocked ER or EGFR receptors (MCF7/ER− and MDA-MB-231/EGFR−, respectively). The only modification of the cell cycle was observed in MCF7/ER− cells after treatment with all of the studied compounds, which suggested the arrest of MCF7 cells in the S-phase after treatment with all of the compounds in all concentration ranges, implying replication repression. Consequently, a decrease in the G2/M-phase as well as an increase in the G0/G1 phase were observed, which also implies the limitation of replication [51]. At the same time, no significant alterations in the MDA-MB-231/EGFR− cell cycle were observed, however, the proapoptotic activity of HIMOXOL observed earlier (in EGFR+ cells [30]) was diminished. Due to previous reports concerning the contribution of OA derivatives to autophagy [30], some further experiments were designed.

Autophagy is responsible for the balance between the production and degradation of cell structures but can lead to the destruction of a cell [52]. Consequently, this process allows organisms to develop properly and maintain homeostasis. Deregulation of autophagy leads to certain disorders, such as the development of neurodegenerative diseases, liver disease, cardiomyopathy, as well as tumor formation [53]. Autophagy can be perceived from a dual perspective. Its inhibition was reported to promote carcinogenesis in cells with a genome subjected to accumulated reactive oxygen species [54]. On the other hand, autophagy induction was shown to provide tumor cell survival in unfavorable conditions, such as insufficient nutrient supply, hypoxia, acidosis, or chemotherapeutic agent exposition [55]. The main autophagy regulators are mTOR, BECN1, MAPLC3, and p62. The kinase mTOR (mammalian target of rapamycin) consists of two distinct complexes, mTOR complex 1 and 2 (mTORC1 and mTORC2), defined by the presence of the key accessory proteins Raptor and Rictor [56]. These two distinct complexes differ in substrate specificity, their upstream regulatory cues, and their localization in the cell. mTOR drives most anabolic processes in the cell, including protein, lipid, cholesterol, and nucleotide synthesis, while it simultaneously increases extracellular nutrient uptake and blocks autophagic catabolism. Thus, it is called an autophagy inhibitor [57].

In our study, no effect of OA on autophagy induction in MCF7/ER− and MDA-MB-231/EGFR− breast cancer cells was observed. Only Br-HIMOLID in the higher compound concentration (i.e., 1.5 × IC_50_) induced autophagy that was manifested by LC3-II protein level induction, which is a crucial marker of autophagy [58]. However, the proautophagic effect of HIMOXOL (observed in both ER+ and EGFR+ cells) was diminished. This could suggest the contribution of ER and EGFR to autophagy induction in MCF7 and MDA-MB-231 cells treated with oleanolic acid semisynthetic derivatives.

Cell adhesion/migration is a complex and dynamic multi-step process that involves a balance between the assembly and disassembly of matrix–cell adhesion sites [59]. However, dysregulation of this balance and the ability of metastasis are crucial features of cancer cells [60]. Thus, using drugs that target adhesion and migration signaling pathways in these cells is one of the most effective strategies in cancer treatment. Proteins that are particularly important in these signaling pathways are, among others, integrins, focal adhesion kinase (FAK), and paxillin [61]. We showed that blocking ER- or EGFR-mediated pathways led to a decrease in the anti-migratory potential of oleanolic acid derivatives in MCF7 and MDA-MB-231 cells. In MCF7/ER+ cells and MDA-MB-231/EGFR+, HIMOXOL and Br-HIMOLID significantly reduced the migratory potential of these cells, which was verified using a wound healing assay [32]. In MCF7 ER− and MDA-MB-231 EGFR− cells only a slight anti-migratory effect of these derivatives was observed.

Immunoidentification of focal adhesion (FA) proteins showed alterations in Tyr397FAK, FAK, and paxillin protein levels, which indicated their role in the response of breast cancer cells to the studied compounds. However, we could not see any alterations in the level of integrin β1, which is one of the key cell adhesion and migration mediators [62]. This could imply a disruption in signal transduction through integrin β1 associated with reduced ER and EGFR status in the studied breast cancer cells.

In turn, a reduction in FAK phosphorylation and total FAK levels in studied cells was observed. Importantly, these proteins are not only associated with adhesion but also with other processes, such as proliferation and survival, angiogenesis, or inflammatory response [63,64]. Thus, evaluation of the role of OA and its derivatives in cancer metabolism requires further detailed studies.

The results presented in this manuscript comprise an assessment of the contribution of OA and its derivatives (HIMOXOL and Br-HIMOLID) to chemosensitivity, genotoxicity, cell cycle, apoptosis, autophagy, and migration of MCF7 and MDA-MB-231 cells concerning the status of ER and EGFR in these cells.

In our studies, we used the parental compound (OA) modified in positions C3, C11, C12, and C28 of its structure. These modifications were based on incorporating individual chemical residues into the respective sites: ketone, hydroxyimine, bromine, and lactone groups and carboxylic group esterification. It was proved that the introduction of a hydroxyimino group at C3 of the oleanan could affect the inhibition of tumor cell proliferation, and additionally cause cell death by apoptosis, autophagy, or necrosis [65]. However, it was reported that the imino group might have diverse biological activities, including antimicrobial activity, antiviral activity, and most importantly, antitumor activity [66]. It is also known that a hydrogen NHOH group may form a weak hydrogen bond with the oxygen atom of the keto- group, and the whole structure of the molecule acquires greater lipophilicity, which allows for easier penetration of the compound into the cell structures [67]. Moreover, the introduction of the lactone to a compound structure increases anticancer activity [68,69]. Additionally, the introduction of bromine improves lipophilicity and biomedical application [70], and compound esterification increases water solubility [71].

Our findings seem to indicate the direction of further chemical modification of the parental OA structure, which should be the incorporation of a hydroxyimino moiety in the C3 position, a bromine atom in position C12, and a lactone residue between C13 and C28 of the OA structure.

Regarding the observed results, we strongly suggest that both receptors may significantly alter the response of cancer cells to studied triterpenoids. This is of high importance, since both receptors are known to affect key cellular processes such as proliferation, apoptosis, autophagy, and migration. Specifically, it was shown that ERα-expressing cells show higher autophagic activity than cells lacking ER expression [72], or that EGFR signaling induces autophagy [73]. Thus, targeting these receptors remains a promising strategy in fighting cancer. Possibly, such therapy might be supported by the administration of certain triterpenoids that should strengthen the therapeutic effect and patients’ outcomes. We showed that modification of the parental compound, i.e., OA, could significantly improve the biological activity of this compound and that the effect is associated with critical metabolic processes of cancer cells. Even if the mechanism is not fully understood, it can be used as crucial information that indicates the direction for designing new compounds with higher biological activity that could be used in different molecular subtypes in fighting breast cancer.

## 4. Materials and Methods

### 4.1. Compounds and Reagents

Oleanolic acid was isolated from an industrial by-product obtained during the process of mistletoe herb essence production. Spectral data of the resulting chemicals were consistent with the data from the literature [74,75,76,77]. The semisynthetic OA derivatives, 12α-bromo-3-hydroxyimonoolean-28→13-olide (Br-HIMOLID) and methyl 3-hydroxyimino-11-oxoolean-12-en-28-oate (HIMOXOL), were synthesized as described in the earlier studies [61,62,63] at the Department of Organic Chemistry, Poznan University of Medical Sciences, Poland (Figure 8). Before use in the experiment, the derivatives were dissolved in DMSO and stored at 4 °C. In all the experiments, the applied range of concentrations was prepared corresponding to the values specified for the IC_50_.

The cell culture media, RPMI 1640, and fetal bovine serum were purchased from Biochrome (PAA, Pashing, Austria). The RPMI 1640 culture medium without phenol red was purchased from Gibco (Thermo Fisher Scientific, Waltham, MA, USA). The MTT, fulvestrant, rapamycin, and camptothecin were obtained from Sigma-Aldrich (Sigma-Aldrich, Munich, Germany). Gefitinib was obtained from Cell Signaling Technology (Danvers, MA, USA).

A crucial reagent in the study of steroid hormones is a dextran-treated charcoal-stripped serum (CSS). Charcoal treatment of serum decreases the concentration of a wide range of peptides and small molecules, e.g., estrogens, growth factors, and cytokines [78]. Due to that, phenol red reveals estrogenic activity, and for MCF7 cells culture medium without this compound was used [79].

### 4.2. Cell Line and Cell Culture

MCF7 (ER+, PR+, EGFR−) and MDA-MB-231 (ER−, PR−, EGFR+) human breast cancer cell lines were obtained from the American Type Culture Collection (HTB-22 and HTB-26, respectively). MDA-MB-231 cells were maintained in RPMI-1640 medium supplemented with 10% fetal bovine serum. MCF7 cells were maintained in RPMI-1640 without phenol red with 5% charcoal-stripped FBS. The cells were cultured in 5% CO_2_ at 37 °C and with saturated humidity. The cell culture medium was changed every 3–4 days.

#### 4.2.1. ER Inhibition with Fulvestrant

Fulvestrant is a representative pure antiestrogen and a Selective Estrogen Receptor Down-regulator (SERD) that causes proteasomal degradation of ERα protein, shutting down the estrogen signaling to induce proliferation arrest [79]. To inhibit estrogen receptor levels, 5 × 10^5^ growing cells were treated with 0.5 µM fulvestrant for 24 h (established experimentally).

#### 4.2.2. EGFR Inhibition with Gefitinib

Gefitinib is a selective epidermal growth factor receptor tyrosine kinase inhibitor [80]. To inhibit EGF receptor level, 5 × 10^5^ growing cells were treated with 20 µM of gefitinib for 72 h (established experimentally).

### 4.3. Viability Assay

Into each well of the 96-well plates, 2.5 × 10^4^/mL growing cells were seeded and compounds were added at the concentration range of 0.5–50 µmol/l. The solvent, DMSO at a concentration of 0.28%, was also applied as a control (Sigma-Aldrich, St. Louis, MO, USA). Two biological duplicates with three technical repeats were created for each concentration with a total volume of 200 µL per well. Then, 20 μL of MTT solution (5 mg/mL) was added to each well. The plates were incubated at 37 °C for 4 h followed by the addition of 100 μL solubilization buffer (10% SDS in 0.01 M HCl). Cell viability was quantified spectrophotometrically using a Labsystems Multiscan RC (Thermo, Champaign, IL, USA). Each experiment was repeated two times, IC50 values were calculated using CampuSyn software version 2022 (ComboSyn Inc., Parammus, NJ, USA), and the standard deviation was calculated using Microsoft Excel software (Microsoft, Redmond, WA, USA).

### 4.4. Colony-Formation Assay

For the colony-forming assay, cells were plated in 6-well plates at 300 cells per plate and were allowed to adhere for 24 h. They were then treated with studied compounds in three different concentrations, corresponding to 0.5 × IC_50_, 1 × IC_50_, and 1.5 × IC_50_ for 24 h. After specified time intervals, the compound-containing media were replaced with fresh, complete media, and cells were grown for ten days with one media change on the fourth day. The colonies formed were fixed with 4% formaldehyde (37 °C; 10 min) and stained with crystal violet (0.5% (*w*/*v*); 1 h, 25 °C). The wells were then washed with distilled water, air-dried, and the colonies were enumerated. The experiment was repeated two times.

### 4.5. Cell Cycle Analysis by Flow Cytometry

MCF7 and MDA-MB-231 cells were treated with indicated OA, Br-HIMOLID, or HIMOXOL concentrations for 24 h. The cells were collected using 0.25% trypsin (Sigma-Aldrich, St. Louis, MO, USA), then washed and resuspended in 100 µL PBS containing 50 µg/mL propidium iodide and 25 µL of ribonuclease A (10 mg/mL) (Sigma-Aldrich, St. Louis, MO, USA). After 1 h of incubation, flow cytometry analysis was performed (FACScan, Becton Dickinson, Franklin Lakes, NJ, USA). The percentages of the cell population in the subphases G0/G1, S, and G2/M (and apoptosis) were calculated from the histograms. The experiment was repeated three times.

### 4.6. Immunodetection

MCF7 and MDA-MB-231 cells were treated for 24 h with three concentrations of OA, Br-HIMOLID, or HIMOXOL, corresponding to 0.5 × IC_50_, 1 × IC_50,_ or 1.5 × IC_50_ (for assessment of proteins involved in adhesion/migration process, sub-cytotoxic concentration of studied compounds causing viability of cells around 60–70% corresponding to 0.5 × IC_50_ was applied) (Table 1). Whole-cell extracts were prepared using a modified RIPA lysis buffer (50 mM Tris–HCl, pH 8.0, 150 mM NaCl, 1% NP40, 0.1% SDS, 100 mM PMSF, 25 μg/mL Na3VO4, 25 μg/mL NaF, 25 μg/mL leupeptin, and 25 μg/mL aprotinin). The protein concentration was measured using a Bradford assay (Sigma-Aldrich, St. Louis, MO, USA), and 40 μg of each extract was loaded onto SDS-PAGE gels. Western blotting was performed according to the standard procedure using a PVDF membrane (Pierce Biotechnology, Rockford, IL, USA) [70]. The following antibodies were used for detection: anti-ERα, anti-EGFR, anti-MAPLC3, anti-mTOR, anti-BECN1, anti-p62, anti-integrin β1, anti-paxillin, anti-pFAK Tyr-397, anti-FAK (Cell Signaling Technology, Danvers, MA, USA), and anti-GAPDH (Santa Cruz Biotechnology, Dallas, CA, USA); 1 μg/mL of each primary antibody was used in the blotting solution. The proteins were visualized using the SuperSignal West Pico Chemiluminescent Substrate (Pierce Biotechnology, Rockford, IL, USA). The optical density (Arbitrary Units) of the bands was measured using the VisionWorks software (NVIDIA, Santa Clara, CA, USA). Representatives of the two experiments are shown in Figure 1, Figure 5 and Figure 7.

### 4.7. Wound Healing Assay

The anti-migratory effects of OA, Br-HIMOLID, and HIMOXOL in human breast cancer cells (6 × 10^5^) were examined by wound healing assay, as described in an earlier report [81]. The cells were seeded in 6-well plates. After the cells reached confluency, an artificial scratch was made with a tip in each of the wells. The cells were washed with PBS and images of the experimental group (cells treated with the concentration of compounds corresponding to 0.5 × IC_50_) were taken at 24 h after treatment with OA, Br-HIMOLID, or HIMOXOL, and compared with control cell data for quantification of cell migration ratio. Each experiment was repeated two times.

### 4.8. Statistical Analysis

The data are means from two separate experiments unless otherwise specified. Statistical analysis was performed by one-way ANOVA (GraphPad Prism 5, San Diego, CA, USA). A *p*-value < 0.05 was considered to be indicative of a significant difference.

## Figures and Tables

**Figure 1 ijms-24-05099-f001:**
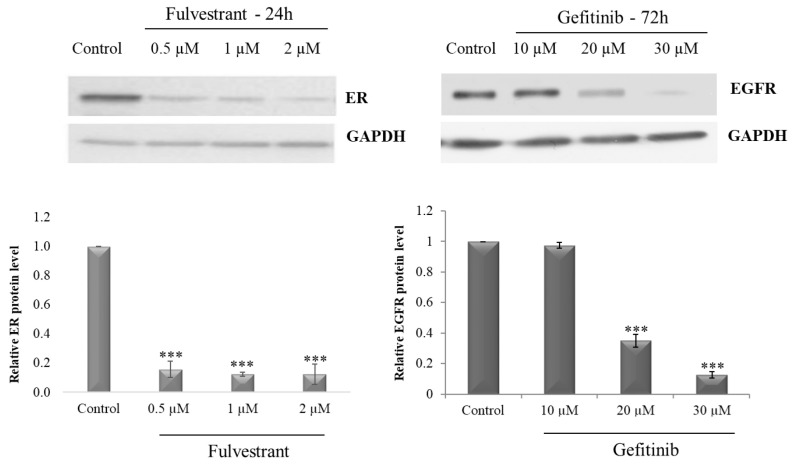
Immunoidentification of estrogen and epidermal growth factor receptors after treatment of studied breast cancer cells with different concentrations of specific receptor blockers in MCF7 and MDA-MB-231 cells. MCF7 cells were treated with fulvestrant in the concentration range 0.5–2 µM for 24 h and MDA-MB-231 cells were treated with gefitinib for 72 h in the concentration range 10–30 µM. GAPDH was used as a loading control. *** *p* < 0.01, relative to control cells.

**Figure 2 ijms-24-05099-f002:**
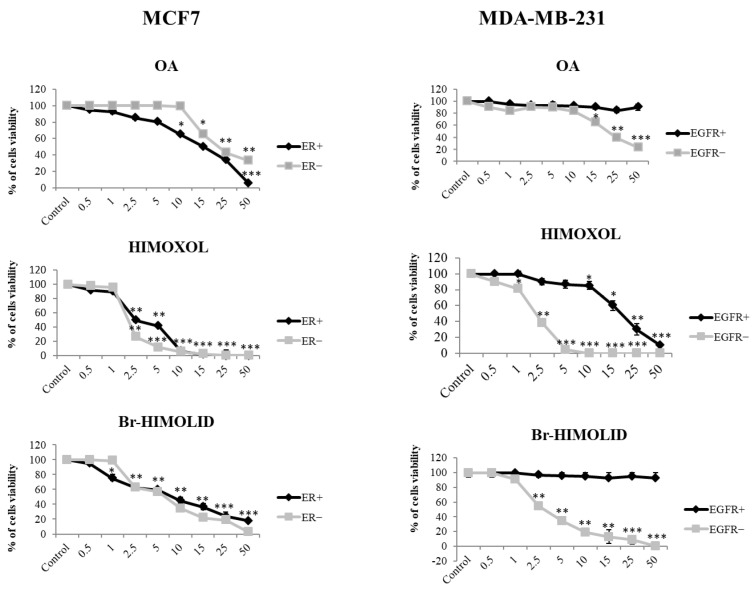
Viability of MCF7 and MDA-MB-231 cells with and without blocking the ER- or EGFR-mediated pathways. Cells were treated with OA, HIMOXOL, or Br-HIMOLID after blocking specific receptors (fulvestrant or gefitinib, respectively), in a concentration range of the compounds 0.5–50 µM for 24 h. Experiments were performed in duplicates. The data from the viability of MCF7/ER+ and MDA-MB-231/EGFR+ cells were published earlier [25,26]. * *p* < 0.05, ** *p* < 0.005, *** *p* < 0.001, relative to control.

**Figure 3 ijms-24-05099-f003:**
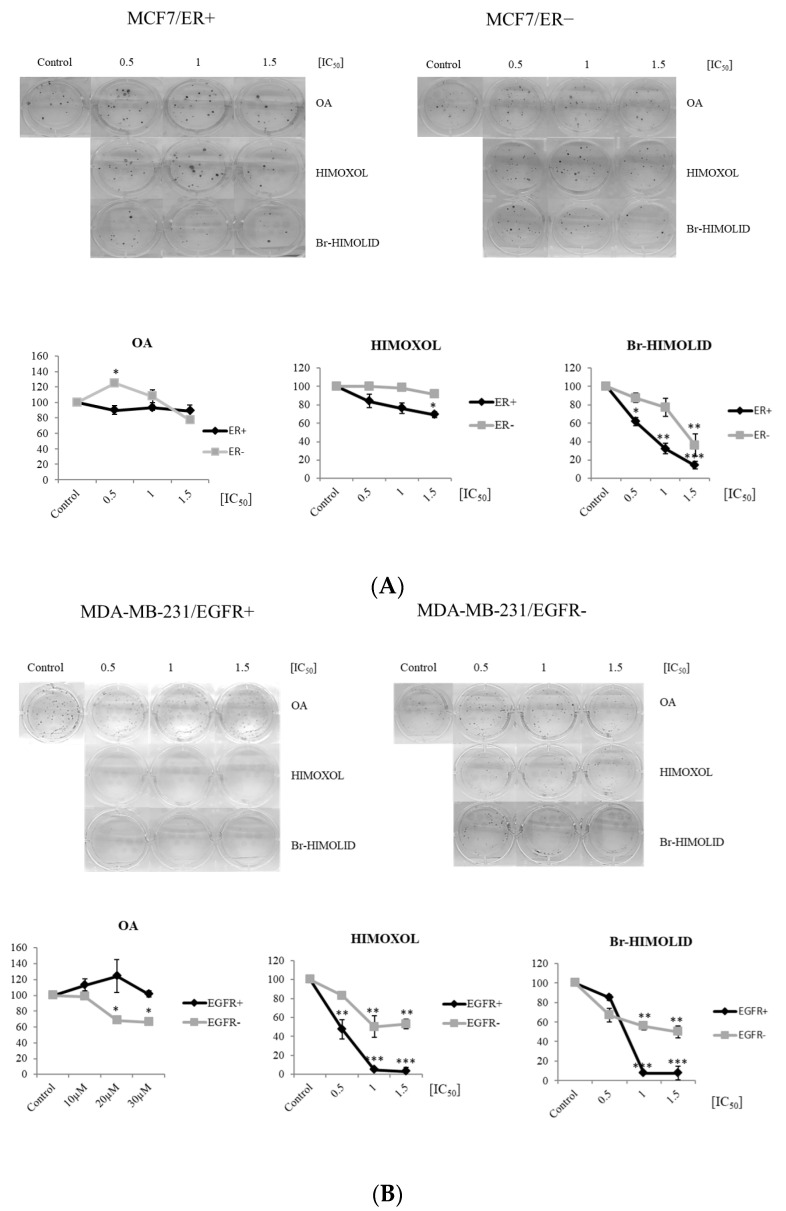
Influence of OA and its derivatives on colony formation potential of breast cancer cells. Breast cancer cells MCF7/ER+ and MCF7/ER− (**A**) as well as MDA-MB-231/EGFR+ and MDA-MB-231/EGFR− cells (**B**) were treated with different concentrations of studied compounds, i.e., OA, HIMOXOL, or Br-HIMOLID, in concentration range corresponding to 0.5, 1, and 1.5 × IC_50_. All experiments were performed in duplicates ± SD. * *p* < 0.05; ** *p* < 0.005; *** *p* < 0.001.

**Figure 4 ijms-24-05099-f004:**
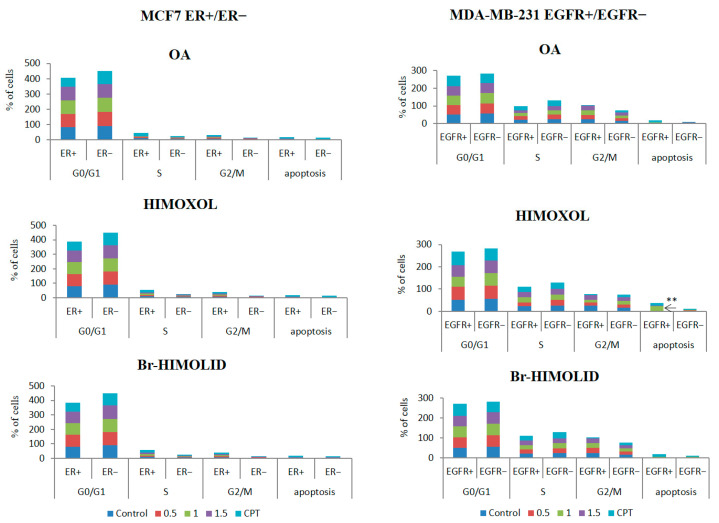
Evaluation of cell cycle modulation and proapoptotic potential of OA or its derivatives. MCF7 and MDA-MB-231 cells (ER+/− and EGFR+/−, respectively) were treated with studied compounds and 0.5×, 1×, or 1.5 × IC_50_ values were applied. Camptothecin (CPT, 5 µM for MCF7 and 20 µM for MDA-MB-231) was used as a positive control. The mean value of three experiments ± SD is shown: ** *p* < 0.05. Cell cycle cytofluorimetry data analyses for MCF7/ER+ and MDA-MB-231/EGFR+ cells were published before [30,31].

**Figure 5 ijms-24-05099-f005:**
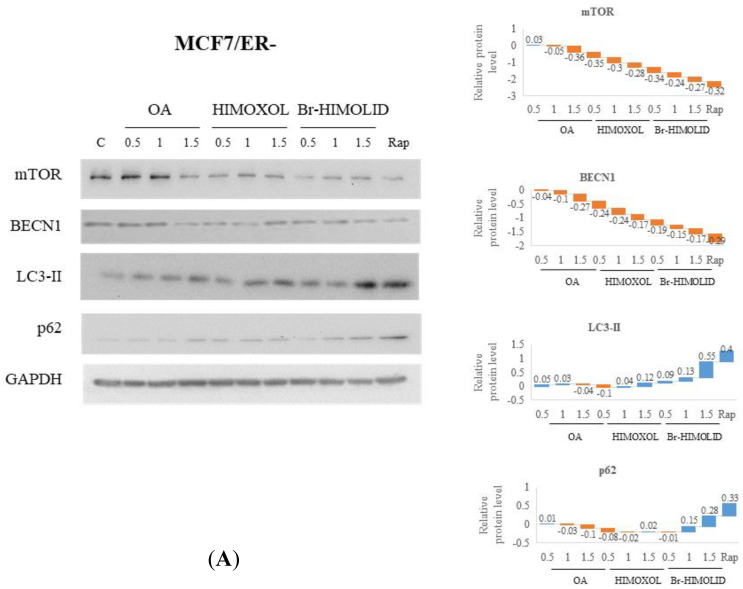
Evaluation of autophagy induction by OA and its derivatives in MCF7/ER− (**A**) and MDA-MB-231/EGFR− (**B**) breast cancer cells. Cells were treated with studied compounds, i.e., 0.5×, 1×, or 1.5 × IC_50_ values for 24 h. Rapamycin (Rap, 100 nM) was used as a positive control. The diagrams present a semiquantitative assessment of target protein levels relative (fold change) to control, untreated samples.

**Figure 6 ijms-24-05099-f006:**
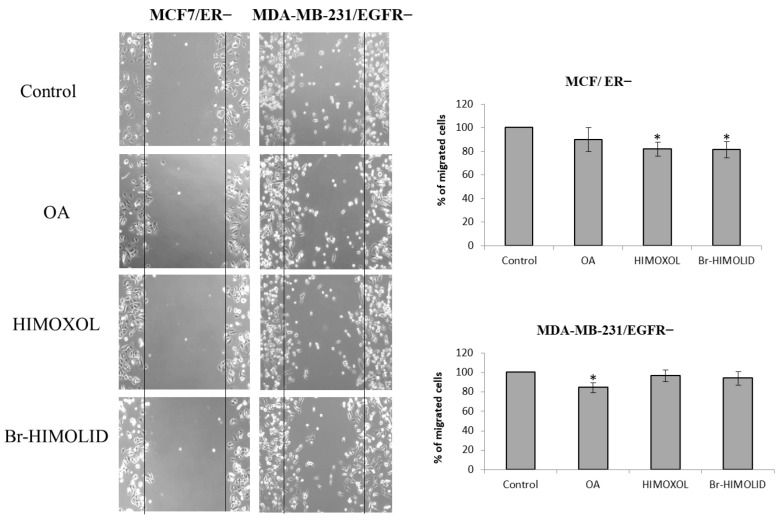
Effect of OA and its derivatives, HIMOXOL and Br-HIMOLID, on wound healing in MCF7/ER− and MDA-MB-231/EGFR− breast cancer cells. Cells were scraped with a pipette tip and treated with OA or OA derivatives (0.5 × IC_50_) for 24 h. The photos represent cell migration under the microscope at 100× magnification after scratch. The migration of studied breast cancer cells was quantified by measuring wound closure cells relative to control, untreated cells. The experiments were repeated two times. * *p* < 0.05 compared to the control cells.

**Figure 7 ijms-24-05099-f007:**
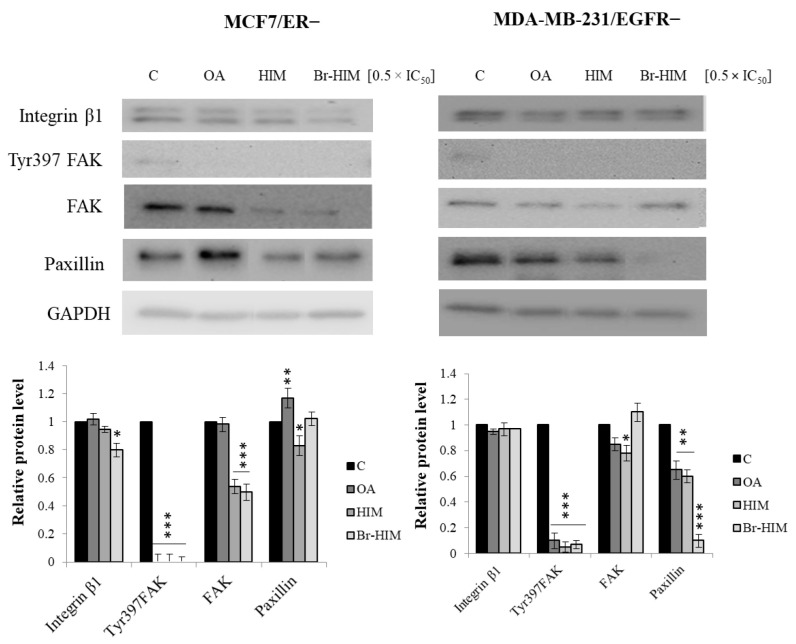
Alterations in proteins mediating adhesion. Evaluation of integrin β1, Tyr397FAK, total FAK, and paxillin levels in MCF7/ER− and MDA-MB-231/EGFR− cells treated with OA or its derivatives for 24 h was performed using Western blot. Data from two independent experiments are shown as mean ± SD. * *p* < 0.05; ** *p* < 0.005; *** *p* < 0.001.

**Figure 8 ijms-24-05099-f008:**
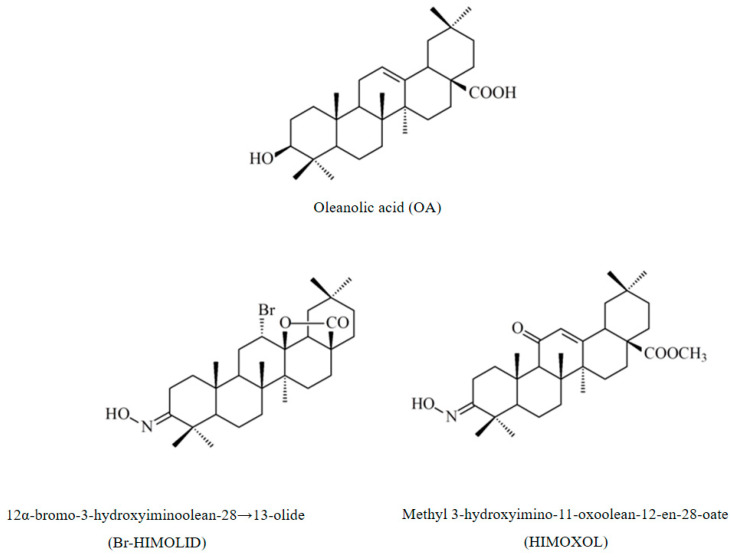
Structure of oleanolic acid (OA) and its derivatives that were studied in breast cancer cells.

**Table 1 ijms-24-05099-t001:** Comparison of OA and cytotoxicity of its two semisynthetic derivatives in MCF7 ER+/ER− and MDA-MB-231 EGFR+/EGFR− cells. The IC_50_ values were calculated based on the concentration–response curves assessed by MTT assay for 24 h. Statistical analysis shows significant differences when the biological effect of studied compounds in ER+/ER− and EGFR+/EGFR− cells was compared: ** *p* < 0.05, *** *p* < 0.001.

		MCF7 IC_50_ [µM]		MDA-MB-231 IC_50_ [µM]	
	Receptor Status	ER+	ER−	EGFR+	EGFR−
Compound					
**OA**		12.7 ± 0.45	31.71 ± 2.98 **	>50	22.47 ± 2.49 **
**HIMOXOL**		3.22 ± 0.42	1.63 ± 0.1 **	21	1.72 ± 0.08 ***
**Br-HIMOLID**		6.58 ± 0.74	3.53 ± 0.4 **	>50	3.44 ± 0.19 ***

## Data Availability

The data are contained within this article.

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
