# Peer review of "Biological Activity of Oleanolic Acid Derivatives HIMOXOL and Br-HIMOLID in Breast Cancer Cells Is Mediated by ER and EGFR"

_ijms, 2023, doi:10.3390/ijms24065099_

Round 1

Reviewer 1 Report

Congratulations to the authors because it is a good work whose purpose is the search for new treatment options for breast cancer with few adverse effects. They present various types of techniques as well as results in two breast cancer cell lines. Even though it's a good job I think it needs some improvements. I am going to make some suggestions; 1. The introduction must be modified so that it follows the same order as the title and the discussion and the meaning of this work. That is, use 3 lines about the current situation of breast cancer to continue with the subject of study of this work, OA and its derivatives and would develop this theme (line 66). Subsequently, to expose the usefulness of the new strategies in two types of breast cancer; estrogen-dependent and triple negative, making a brief description of its receptors. I think that in this way it is more in line with the structure of the current document. 2. They provide many results and it would be easier for them to understand and read a diagram/drawing of the figures. It would be necessary for point 2.5 (drawing in waterfall). 3. The results obtained for OA, especially in the feasibility study, are very striking. A more exhaustive explanation would be interesting. 4. Line 294. Abbreviation of NfKb

Author Response

First, we would like to thank the reviewers for their kind comments and constructive criticism. Please find addressed all comments below and in the text. All changes are highlighted in yellow. The manuscript was edited for proper English language, grammar, punctuation, spelling, and overall style, so we hope it now matches the journal standard.

Reviewer 1

Congratulations to the authors because it is a good work whose purpose is the search for new treatment options for breast cancer with few adverse effects. They present various types of techniques as well as results in two breast cancer cell lines. Even though it's a good job I think it needs some improvements. I am going to make some suggestions;

  1. The introduction must be modified so that it follows the same order as the title and the discussion and the meaning of this work. That is, use 3 lines about the current situation of breast cancer to continue with the subject of study of this work, OA and its derivatives and would develop this theme (line 66). Subsequently, to expose the usefulness of the new strategies in two types of breast cancer; estrogen-dependent and triple negative, making a brief description of its receptors. I think that in this way it is more in line with the structure of the current document.

All the suggestions were addressed. Specifically, the manuscript structure was amended and the current situation of breast cancer was updated. Similarly, more detailed description of the model breast cancer cell lines was provided.

  1. They provide many results and it would be easier for them to understand and read a diagram/drawing of the figures. It would be necessary for point 2.5 (drawing in waterfall).

The recommended modification of the diagram in Figure 5 (point 2.5) was applied and we hope it is clear now.

  1. The results obtained for OA, especially in the feasibility study, are very striking. A more exhaustive explanation would be interesting.

The section regarding the compounds, their structure and association between modifications and expected/observed effects was elaborated.

  1. Line 294. Abbreviation of NfKb

It was modified.

First, we would like to thank the reviewers for their kind comments and constructive criticism. Please find addressed all comments below and in the text. All changes are highlighted in yellow. The manuscript was edited for proper English language, grammar, punctuation, spelling, and overall style, so we hope it now matches the journal standard.

Reviewer 1

Congratulations to the authors because it is a good work whose purpose is the search for new treatment options for breast cancer with few adverse effects. They present various types of techniques as well as results in two breast cancer cell lines. Even though it's a good job I think it needs some improvements. I am going to make some suggestions;

  1. The introduction must be modified so that it follows the same order as the title and the discussion and the meaning of this work. That is, use 3 lines about the current situation of breast cancer to continue with the subject of study of this work, OA and its derivatives and would develop this theme (line 66). Subsequently, to expose the usefulness of the new strategies in two types of breast cancer; estrogen-dependent and triple negative, making a brief description of its receptors. I think that in this way it is more in line with the structure of the current document.

All the suggestions were addressed. Specifically, the manuscript structure was amended and the current situation of breast cancer was updated. Similarly, more detailed description of the model breast cancer cell lines was provided.

  1. They provide many results and it would be easier for them to understand and read a diagram/drawing of the figures. It would be necessary for point 2.5 (drawing in waterfall).

The recommended modification of the diagram in Figure 5 (point 2.5) was applied and we hope it is clear now.

  1. The results obtained for OA, especially in the feasibility study, are very striking. A more exhaustive explanation would be interesting.

The section regarding the compounds, their structure and association between modifications and expected/observed effects was elaborated.

  1. Line 294. Abbreviation of NfKb

It was modified.

Reviewer 2 Report

This study mainly addresses the effects of oleanolic acid semisynthetic derivatives on apoptosis, autophagy or diminishing the migratory and invasive potential of breast cancer related to estrogen receptor (ER) and epidermal growth factor receptor (EGFR). This study is comprehensive including lots of in vitro experiments. Authors compared the efficacy of these derivatives to be potential compared to well-known standards, fulvestrant and gefitinib. The effects of compounds were not superior to standards. The references are current and appropriate. The introduction, discussion and conclusion parts are well-presented. Although the results are not promising, this study deserves to be accepted after minor revision due to the well-designed study and very extensive experimental methods.

Here are the comments:

The quality of figures is not good please improve it.

The IC50 must be written in as the IC50 in whole manuscript.

Please add the chemical structures of OA, HIMOXOL and Br-HIMOLID. The authors can discuss how  the chemistry of the oleanolic acid and derivatives affected the activity.

Author Response

First, we would like to thank the reviewers for their kind comments and constructive criticism. Please find addressed all comments below and in the text. All changes are highlighted in yellow. The manuscript was edited for proper English language, grammar, punctuation, spelling, and overall style, so we hope it now matches the journal standard.

This study mainly addresses the effects of oleanolic acid semisynthetic derivatives on apoptosis, autophagy or diminishing the migratory and invasive potential of breast cancer related to estrogen receptor (ER) and epidermal growth factor receptor (EGFR). This study is comprehensive including lots of in vitro experiments. Authors compared the efficacy of these derivatives to be potential compared to well-known standards, fulvestrant and gefitinib. The effects of compounds were not superior to standards. The references are current and appropriate. The introduction, discussion and conclusion parts are well-presented. Although the results are not promising, this study deserves to be accepted after minor revision due to the well-designed study and very extensive experimental methods.

Here are the comments:

  1. The quality of figures is not good please improve it.

The quality of figures was improved. However, we would like to stress that the manuscript graphic form may be significantly affected by the conversion process during the submission.

  1. The IC50 must be written in as the IC50 in whole manuscript.

It was corrected.

  1. Please add the chemical structures of OA, HIMOXOL and Br-HIMOLID. The authors can discuss how the chemistry of the oleanolic acid and derivatives affected the activity.

The chemical structure of studied compounds was included in Figure 8
(Part 4. Materials and Method section, 4.1 Compounds and reagents).

The discussion regarding the association between incorporated chemical modifications to OA structure and their contribution to the biological activity of studied compounds was elaborated.

Round 2

Reviewer 1 Report

I consider that it is correct for its publication.